# Chebulic Acid Prevents Hypoxia Insult via Nrf2/ARE Pathway in Ischemic Stroke

**DOI:** 10.3390/nu14245390

**Published:** 2022-12-19

**Authors:** Rong Zhou, Kuan Lin, Changlong Leng, Mei Zhou, Jing Zhang, Youwei Li, Yujing Liu, Xiansheng Ye, Xiaoli Xu, Binlian Sun, Xiji Shu, Wei Liu

**Affiliations:** 1Wuhan Institutes of Biomedical Sciences, School of Medicine, Jianghan University, Wuhan 430056, China; 2Institute of Cerebrovascular Disease, Jianghan University, Wuhan 430056, China

**Keywords:** ischemic stroke, Chebulic acid, Nrf2, neuroprotection, antioxidants

## Abstract

Excessive reactive oxygen species (ROS) production contributes to brain ischemia/reperfusion (I/R) injury through many mechanisms including inflammation, apoptosis, and cellular necrosis. Chebulic acid (CA) isolated from *Terminalia chebula* has been found to have various biological effects, such as antioxidants. In this study, we investigated the mechanism of the anti-hypoxic neuroprotective effect of CA in vitro and in vivo. The results showed that CA could protect against oxygen-glucose deprivation/reoxygenation (OGD/R) induced neurotoxicity in SH-SY5Y cells, as evidenced by the enhancement of cell viability and improvement of total superoxide dismutase (T-SOD) in SH-SY5Y cells. CA also attenuated OGD/R-induced elevations of malondialdehyde (MDA) and ROS in SH-SY5Y cells. Nuclear factor-E2-related factor 2 (Nrf2) is one of the key regulators of endogenous antioxidant defense. CA acted as antioxidants indirectly by upregulating antioxidant-responsive-element (ARE) and Nrf2 nuclear translocation to relieve OGD/R-induced oxidative damage. Furthermore, the results showed that CA treatment resulted in a significant decrease in ischemic infarct volume and improved performance in the motor ability of mice 24 h after stroke. This study provides a new niche targeting drug to oppose ischemic stroke and reveals the promising potential of CA for the control of ischemic stroke in humans.

## 1. Introduction

Stroke is one of the leading causes of death and disability in the global population, placing a heavy burden on families and society. Ischemic stroke accounts for 60~70% of stroke cases and is caused by a narrowing or occlusion of the arteries, resulting in insufficient blood and oxygen supply to the brain. Currently, the primary treatments for ischemic stroke are surgery and drug therapy, and the pathological intervention of ischemic injury is focused on improving cerebral blood circulation and neuroprotection [1]. Although significant progress has been made in the treatment of stroke, there are still many deficiencies in the treatment methods and effects [2]. For example, intravenous recombinant tissue plasminogen activator (R-TPA) is currently approved for the clinical treatment of ischemic stroke [3]; however, most patients do not have access to thrombolytic therapy because of the three-hour treatment window. Therefore, it is of great importance to find new drugs that can prevent stroke and reduce its poor prognosis.

Currently, drug discovery based on a single target is unsatisfactory, and research and development of new drugs from the perspective of multiple targets are increasingly recognized by researchers in the field of medicine worldwide [4,5]. Traditional Chinese medicines (TCMs) have been used to treat human diseases for thousands of years. The unique advantage of TCMs is that they are characterized by multiple active components that can act on multiple targets in the treatment of diseases. Thus, it is an effective approach to research drugs utilized in traditional Chinese medicine for the treatment of ischemic stroke.

Numerous studies have shown that oxidative stress plays a key role in ischemic brain injury [6]. When cerebral ischemia and reperfusion occur, a harmful cycle of inflammation, oxidative stress, and apoptosis occur, ultimately leading to neuronal cell death. Under normal circumstances, the production and clearance of oxygen-free radicals in the body are balanced. During cerebral ischemia, especially during reperfusion, the production of free radicals exceeds the scavenging capacity to activate a series of inflammatory markers in ischemic areas, which can directly damage cells and induce cell necrosis and apoptosis, a process known as oxidative stress injury [7,8,9]. Nrf2 is an important regulatory factor in endogenous antioxidant defense [10], which can promote the transcription of a variety of antioxidant genes to reduce oxidative stress damage [11,12].

*Terminalia chebula,* a traditional medicine widely used in Asia to treat various diseases, is the dried ripe fruit of *Terminalia chebula* Retz. tomentella Kurt. It contains a variety of bioactive components such as tannins, organic acids, triterpenes, and CA, which have been reported to have strong antioxidant and anticancer effects [13].

Although CA has various biological effects, the specific neuroprotective effects and mechanisms of CA in ischemic stroke have not been reported. Therefore, in this study, we investigated the neuroprotective effects of CA on ischemic stroke both in vitro and in vivo and examine its related mechanisms.

## 2. Materials and Methods

### 2.1. Cell Culture

SH-SY5Y cells were obtained from Procell Life Science and Technology (Wuhan, China). Cells were cultured in Dulbecco’s modified Eagle’s medium (DMEM)/F12 (1:1) (Gibco) supplemented with 10% fetal bovine serum. For OGD/R model, the glucose-containing medium was first removed and the cells were cleaned with PBS, followed by the addition of glucose-free DMEM to the cells. The cells were then exposed to hypoxia (37 °C, 94% N_2_, 5% CO_2_, 1% O_2_) for 5 h. After that, Oxygen-glucose deprivation was then terminated and cells were cultured under normal conditions (37 °C, 95% air, 5% CO_2_) for 2 h.

### 2.2. CA Preparation and Treatment

CA (Biopurify, Chengdu, China) was first dissolved in DMSO (Sigma, St. Louis, MO, USA) to a stock solution at a concentration of 60 µg/µL. It was further diluted with DMEM or glucose-free DMEM and sterile filtered using a 0.22 μm filter before being added to the SH-SY5Y cells. As mentioned above, the OGD/R model was constructed, and the cells were divided into different groups according to the experimental requirements. The final concentration of DMSO was 0.05% (*v/v*).

### 2.3. CCK8 Assay

The CCK8 assay (Beyotime, Shanghai, China) was used to evaluate cell viability. For the CA toxicity test, cells were inoculated into 96-well plates (10^4^/mL) with CA (10–300 µg/mL), and cells without CA were used as the positive control. For the detection of CA protection, cells were divided into the following groups: (1) normal control groups, (2) OGD/R groups, and (3) OGD/R+ CA (10–100 µg/mL) groups. After 24 h, 10 µL of CCK8 was added to each well and then incubated for 1 h. Absorbance at 450 nm was measured with a microplate reader.

### 2.4. CA Antioxidant Activity

SH-SY5Y cells were grouped as follows: (1) normal control, (2) OGD/R, (3) 10 μg/mL CA + OGD/R, (4) 20 μg/mL CA + OGD/R, and (5) 40 μg/mL CA + OGD/R. Total superoxide dismutase (T-SOD) activity and lipid peroxidation levels were determined with a WST-8 and MDA assay kit (Beyotime, Shanghai, China).

### 2.5. Immunofluorescence of the ROS

A 2′,7′-dichlorofluorescein diacetate (DCFH-DA) was prepared in a serum-free medium using a ROS detection kit (Beyotime, Shanghai, China). Cells in each group were collected and suspended in DCFH-DA solution for 20 min. DAPI was dropped, sealed, and observed under a forward microscope (DP72, Olympus, Tokyo, Japan).

### 2.6. Detection of the ROS and Apoptosis Rate by Flow Cytometry

DCFH-DA was diluted with serum-free medium at 1:1000. After modeling, the medium was removed, and 500 µL of diluted DCFH-DA was added. After incubating for 20 min in the cell culture box at 37 °C, ROS content was determined using flow cytometry.

Cell apoptosis rate was detected with a Cell Apoptosis Kit (Beyotime, Shanghai, China). Cells were first digested with trypsin, and then obtained cell precipitates. The cell precipitates were resuspended in a mixture of binding buffer and FITC-labeled annexin V, and then incubated at room temperature for 10 min. After 10 μL propidium iodide was added and incubated on ice for 10 min, apoptotic cells were measured by FACS analysis.

### 2.7. Immunofluorescence of Nrf2 Nuclear Translocation

Cells were seeded in a cell slide (JinAn Biological, Shanghai, China). After modeling, 300 µL 4% pre-cooled paraformaldehyde was added to each well and fixed for 20 min. The cells were then sealed with 5% BSA. They rested for 40 min, then cleaned with PBS. Nrf2 antibody (1:500) was used to incubate at 4 °C overnight. After adding fluorescent secondary antibody (1:200), incubated at 37 °C for 1 h. TBST was cleaned three times for 5 min, the cell culture slide was removed, DAPI was dropped, and observed under a forward microscope.

### 2.8. Animals and Middle Cerebral Artery Occlusion (MCAO) Model Establishment

C57BL/6 mice were obtained from the Laboratory Animal Center of Yangzhou University (Yangzhou, China; certificate number SCXK, 2017–0007). The MCAO model was established, and the mice were first anesthetized. After the skin was disinfected, it was incised; the carotid arteries were separated by blunt dissection. A silk suture was advanced from the external carotid artery to the internal carotid artery. Only mice with a ≥60% reduction in cerebral blood flow were included in this study. For mice in the sham group, only external carotid artery was separated, and no suture was placed.

### 2.9. Animal Experiments Drug Administration

The mice were randomly divided into six groups: (1) sham (control group); (2) MCAO group; (3) 10 mg/kg CA + MCAO group; (4) 20 mg/kg CA + MCAO group; (5) 40 mg/kg CA + MCAO group; (6) 80 mg/kg CA + MCAO group. Each group received intraperitoneal injection of different concentrations of the CA or vehicle (PBS), and CA was first dissolved in DMSO to a stock at a concentration of 60 µg/mL, and was further diluted with PBS and sterile filtered using a 0.22 μm filter before intraperitoneal injection. The final concentration of DMSO was 2% (*v/v*). Vehicle or CA was administered once daily from three days before modeling, and then given again 1.5 h after modeling on the fourth day. The intraperitoneal injection concentration was 0.1 mL/10 mg. All experiments began 24 h after the MCAO.

### 2.10. Behavioral Tests

The rotarod test was used to evaluate balance and motor coordination in mice. Before surgery, all mice received three days of pre-training. The time taken for each mouse to walk on the rotarod was recorded and repeated three times for each mouse. Finally, the average time value was taken.

The modified neurological severity score (mNSS) was used to evaluate the level of neurological deficits in mice, including motion, sensation, beam balance, and reflection/abnormal motion. With higher scores indicating greater damage. The score for all mice was obtained 24 h after ischemia treatment.

#### 2.11. 2,3,5-Triphenyl-tetrazolium Chloride (TTC) Staining

The mice were decapitated 24 h after MCAO, anesthetized with 10% chloral hydrate, and the volume of cerebral infarction was determined. Immediately after the brain tissue was removed, it was placed in a freezer at −20 °C for 20 min, cut into 2 mm thick slices, and incubated in 2% TTC (Sigma, USA) 37 °C solution for 15–30 min. The infarct volume was then calculated.

### 2.12. TUNEL Staining

Brain tissue was embedded in paraffin, then cut into 5 μm slices and baked for 2 h. After brain tissue was dewaxed and hydrated, 20 μg/mL DNase-free protease K (Beyotime, Shanghai, China) was added for 15 min, and 50 μL TUNEL assay solution was added for 60 min, washed with PBS three times. The red signal of the apoptotic cells was observed using a fluorescence microscope.

### 2.13. Western Blot Experiment

Cell particles were lysed with RIPA lysate. Proteins were normalized to 30 μg per lane, resolved on 10% polyacrylamide gels, and subsequently blotted onto PVDF membranes. The membrane was placed in 5% (*w/v*) skim milk for 1 h, it was then incubated at 4 °C overnight with primary antibodies (all at 1:1000), followed by 1 h at room temperature with secondary antibodies. The bands were visualized using enhanced chemiluminescence and quantified using densitometry. All blots presented are representative of at least three independent experiments.

### 2.14. Statistical Analysis

All results were expressed as mean ± SD. One-way analysis of variance (ANOVA) was used for statistical analysis. All experiments were repeated at least three times, and *p* ˂ 0.05 was considered statistically significant.

## 3. Results

### 3.1. Cytotoxicity of CA and Its Protective Effects on OGD/R-Induced SH-SY5Y Cells

The cytotoxic potential of CA (Figure 1A) in SH-SY5Y cells was evaluated using a CCK8 assay (Figure 1B). CA showed no cytotoxicity at the concentrations of 10–100 μg/mL (Figure 1B). Meanwhile, CA distinctly increased the survival rate of SH-SY5Y cells treated with OGD/R in a dose-dependent manner compared to the model group (Figure 1C). These results indicate that CA has a wide safe concentration range and has a protective effect on OGD/R-induced cells. We found that the optimal concentration of CA administered to the cells for subsequent experiments was 10–40 μg/mL. Superoxide dismutase (SOD) is an important antioxidant enzyme that catalyzes the disproportionation of superoxide anions to produce hydrogen peroxide and oxygen (O_2_). In our results, enzymatic antioxidant (T-SOD) activity increased 1.3 times in the 40 μg/mL CA group compared with the control group (Figure 1D). Malondialdehyde (MDA) is a natural product of lipid oxidation that is markedly increased in animal or plant cells suffering from oxidative stress. The lipid peroxidation level significantly decreased when the SH-SY5Y cells was CA-treated at a concentration of 20 μg/mL (Figure 1E).

### 3.2. CA Reduced OGD/R-Induced ROS Production and Apoptosis in SH-SY5Y Cells

To investigate the effect of CA on oxidative stress, immunofluorescence assays, and flow cytometry was used to detect the content of ROS in different groups of SH-SY5Y cells. Immunofluorescence assays showed that the production of ROS in SH-SY5Y cells was inhibited in a dose-dependent manner in the CA group compared with that in the OGD/R group (Figure 2A,B). The same trend was observed using flow cytometry (Figure 2C,D).

A series of injuries after ischemia-reperfusion eventually lead to cell apoptosis; therefore, the apoptotic cells in each group of cells were detected by flow cytometry using Annexin V labeled with FITC. The number of apoptotic cells increased 3.4-fold after OGD/R treatment compared to the control group, while CA treatment reduced OGD/R-induced apoptosis in all treatment groups. The number of apoptotic cells decreased to 61.7% in the model group when treated with CA at the concentration of 40 μg/mL (Figure 2E,F).

### 3.3. CA Promotes Nuclear Translocation of Nrf2 in SH-SY5Y Cells Induced by OGD/R

Nrf2 is an important regulatory factor in antioxidant defense. Increasing evidence has confirmed that Nrf2 plays an important role in protecting brain cells from ischemic injury. Nrf2 is a promising therapeutic target for stroke prevention [14]. Therefore, we investigated whether CA induced Nrf2 translocation into the nucleus of SH-SY5Y cells. Immunofluorescence microscopy showed that CA promoted Nrf2 transfer into the nucleus of SH-SY5Y cells after OGD/R induction (Figure 3A,B). Similar results were also reflected in the western blot analysis, which showed that the expression levels of total Nrf2 and Nrf2 in the nucleus in the CA-treated groups were increased compared to the OGD/R group (Figure 3B–D). These results suggest that the antioxidant effects of CA may be related to the activation of proteins in the Nrf2 pathway. It has been shown that the Keap1- Nrf2-ARE signaling pathway is involved in the protective effect of stroke on brain tissue [15]. Nrf2 is released from Kelch-like ech-associated protein 1 (Keap1), and then translocated to the nucleus, where it binds to AREs in the nucleus and regulates the transcription of downstream targets to promote the expression of a variety of antioxidant genes [16], such as heme oxygenase-1 (HO-1) and SODs. During a stroke, excessive nitric oxide produced by inducible nitric oxide synthase (iNOS) contributes to a cascade of inflammation and neuronal death and further worsens primary brain injury [17]. Figure 3E–H shows that CA promoted the expression of the antioxidant factor HO-1 and decreased the expression of iNOS after OGD/R.

### 3.4. CA Decreased Cerebral Damage and Apoptosis Following Ischemia-Reperfusion In Vivo

To investigate whether CA has the same protective effect in vivo, C57 mice were randomly divided into Sham, MCAO, and MCAO + CA groups. Reperfusion was performed for 24 h according to the surgical methods mentioned above (see Figure 4A for schematic diagram). First, neurological deficit score (mNSS) and infarct volume were measured. The results showed that the neurological deficit score in the CA groups was significantly lower than that in the MCAO group (Figure 4B,C). The effect of CA on the apoptosis of brain cells was measured by TUNEL staining. As shown in Figure 4D,E, the number of TUNEL-positive neurons in the MCAO group significantly increased after ischemia-reperfusion injury, whereas CA administration effectively reduced the apoptosis of nerve cells in vivo.

### 3.5. CA Reduced Infarct Volume after Ischemia-Reperfusion In Vivo

TTC staining was used to quantify infarct volume (Figure 5A,B). Normal brain tissue appears scarlet, whereas infarcts appear white in color. After MCAO, the cerebral infarction volume increased significantly, while the cerebral infarction volume in the treatment groups decreased in a quantitative dependence. At a dose of 80 mg/kg, the infarct volume was reduced to 5.9 times that in the MCAO group. The expression of Nrf2, Keap1, HO−1, and iNOS in brain tissue was also observed. Similarly, the expression of antioxidant stress-related proteins was upregulated after CA compared with that in the MCAO group, while iNOS was downregulated (Figure 5C–G).

## 4. Discussion

Stroke is one of the most important diseases that threaten human life and health worldwide [18]. Stroke includes hemorrhagic stroke and ischemic stroke, of which ischemic stroke accounts for 60~70% of all stroke cases [19]. Many studies have shown that the rapid increase of ROS after the occurrence of acute ischemic stroke can cause an imbalance in the body’s antioxidant response, leading to a series of pathophysiological events, including inflammatory response, apoptosis, and autophagy, ultimately leading to neurodegeneration and nerve cell death [20,21].

Ischemic stroke occurs when an artery in the brain is blocked, resulting in insufficient oxygen and glucose supply to the damaged area to maintain normal cellular activity. This in turn leads to changes in the electrochemical and metabolic levels of cells and increases the release of toxic substances [22], one of which is the increased production of ROS [23,24]. Although reperfusion therapy is currently the widely accepted treatment for ischemic stroke, reperfusion also has a significant negative impact on brain cells, including a significant increase in ROS levels [25]. At this point, the body’s antioxidant mechanism cannot cope with the damage caused by oxidative substances, such as reactive oxygen species, which eventually leads to apoptosis and necrosis of nerve cells. 

Studies have shown that CA can reduce cellular damage by reducing the level of ROS and the content of oxidized glutathione [26,27]. Some studies have also shown that CA participates in the antioxidant process by regulating Nrf2 expression in glomerular mesangial cells and hepatocytes [28,29,30]. However, whether CA can also regulate Nrf2 expression in neurocytes to resist oxidative stress injury has rarely been studied.

Many studies have proved that Nrf2 plays a vital role in protecting brain cells from ischemic injury. As our study showed, in vitro, nuclear translocation of Nrf2 in SH-SY5Y cells was not obvious after cerebral ischemia-reperfusion simulation using the OGD/R model. However, the nuclear translocation of Nrf2 increased significantly after treatment with different concentrations of CA, and the same trend was verified by western blot analysis. These findings indicate that CA effectively activates the Nrf2-ARE regulatory element and protects the collective from oxidative stress in an OGD/R mouse model. Similarly, ROS expression in SH-SY5Y cells increased significantly after OGD/R, whereas it decreased in the CA treatment groups.

Inflammatory factors also play critical roles in cerebral I/R injury. In the early stages of cerebral ischemia, a large number of pro-inflammatory factors are produced in the tissues [31]. Meanwhile, the interaction between oxidative stress and neuroinflammation in ischemic stroke has been extensively studied [32,33]. Nerve cell death, damage to the integrity of the blood-brain barrier (BBB), and enlarged cerebral infarction are related to a variety of inflammatory factors triggered by ROS [34]. Our study showed that the brain tissue of mice in the MCAO group expressed more inflammatory factors, which significantly decreased after treatment with CA. However, the exact molecular mechanisms need to be further explored. 

The pathological processes after cerebral ischemia-reperfusion mainly include oxidative stress, excitatory toxicity, intracellular calcium overload, inflammation, and apoptosis, which ultimately determine the size of the infarction [35]. Although the antioxidant effects of CA have been demonstrated, its neuroprotective effects after ischemia-reperfusion have rarely been studied. Our study found that neurological and motor impairment occurred in mice 24 h after MCAO; however, compared with the model group, both impairments were improved in the CA-treated groups, and the high-dose groups showed more significant improvement than the low-dose groups. Similarly, neuronal apoptosis and the size of cerebral infarction after MCAO were improved in mice treated with CA.

In conclusion, in our study, we found that CA promoted the migration of antioxidant regulator Nrf2 from the cytoplasm to the nucleus and that Nrf2 into the nucleus promotes the increased expression of antioxidant proteins such as HO-1and SOD, these antioxidant proteins can resist the damage of cells and brain tissue caused by excessive reactive oxygen species and inflammatory factors such as iNOS produced during ischemia. Our results found that CA has excellent neuroprotective effects and can be used as a promising new drug for the treatment of acute ischemic stroke; however, its mechanism of action needs to be further elucidated.

## Figures and Tables

**Figure 1 nutrients-14-05390-f001:**
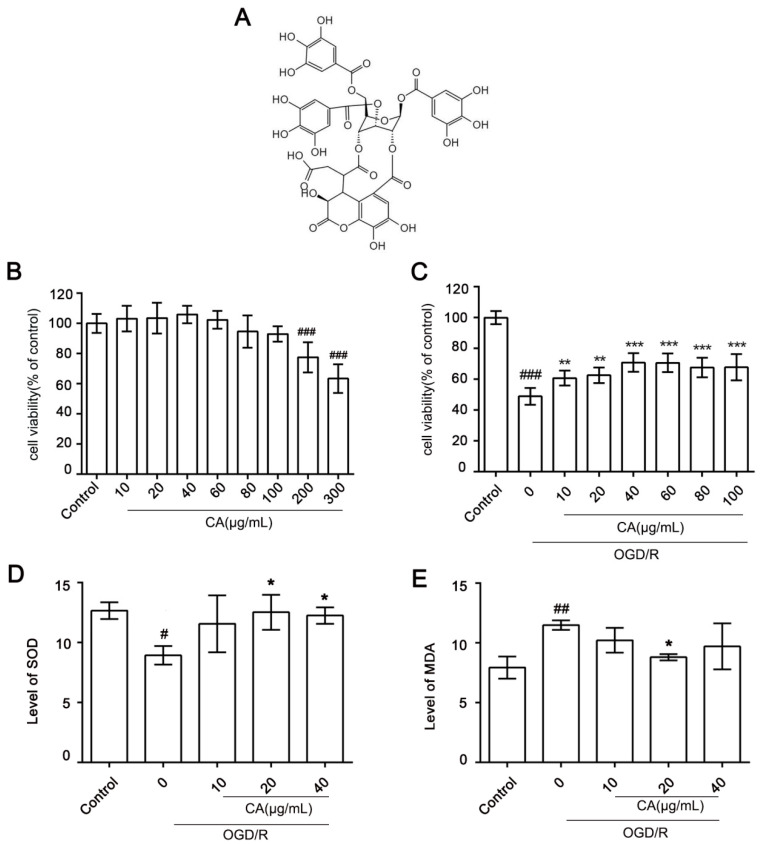
Chebulic acid attenuates OGD/R induced cell death in SH-SY5Y cells. (**A**) Chemical structure of chebulinic acid (CA). (**B**) SH-SY5Y cells were incubated for 24 h with various concentrations of CA (10–300 μg/mL). Cell viability was determined using CCK8 assay; bars represent the percentage of cell viability. (**C**) SH-SY5Y cells were inoculated with different concentrations of CA (10–100 μg /mL) and exposed to OGD/R. (**D**) T-SOD levels in cells. (**E**) MDA levels in cells. Results are expressed as mean ± SD. # *p* < 0.05, ## *p* < 0.01, and ### *p* < 0.001 compared with the control. * *p* < 0.05, ** *p* < 0.01, and *** *p* < 0.001 versus OGD/R group.

**Figure 2 nutrients-14-05390-f002:**
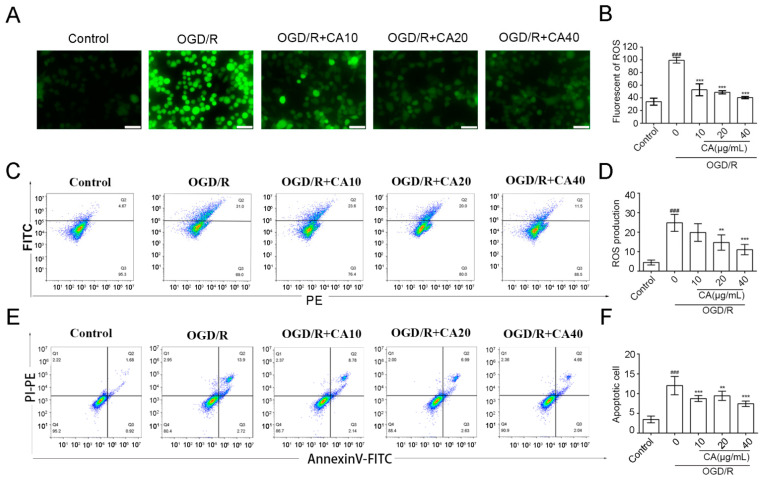
Effect of CA on ROS levels. (**A**,**B**) Representative images of ROS with magnification at 20×. (**C**,**D**) ROS production was detected in each condition. (**E**,**F**) The apoptosis rate was measured using flow cytometry. Results are expressed as mean ± SD. ### *p* < 0.001 compared with the control. ** *p* < 0.01, and *** *p* < 0.001 versus OGD/R group.

**Figure 3 nutrients-14-05390-f003:**
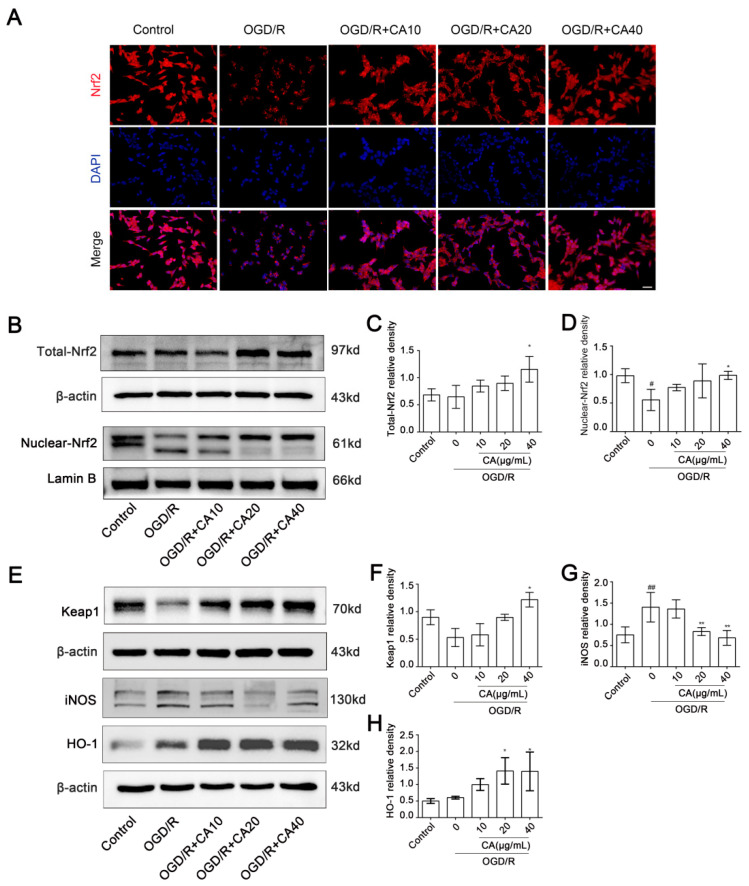
Nrf2 nuclear translocation was enhanced by CA in OGD/R-insulted SH-SY5Y cells. (**A**) SH-SY5Y cells in different groups were first stained with anti- Nrf2 antibody (red) and then counterstained with DAPI (blue) (40×). (**B**–**D**) The expression of nuclear Nrf2 and total Nrf2 were determined by western blot. (**E**–**H**) Chebulinic acid promoted the activation of Nrf2-ARE pathways. Western blot analysis was carried out with specific antibodies for detecting the activation of Keap1, iNOS, HO-1. Results are expressed as mean ± SD. # *p* < 0.05, and ## *p* < 0.01 compared with the control. * *p* < 0.05, and ** *p* < 0.01 versus OGD/R group.

**Figure 4 nutrients-14-05390-f004:**
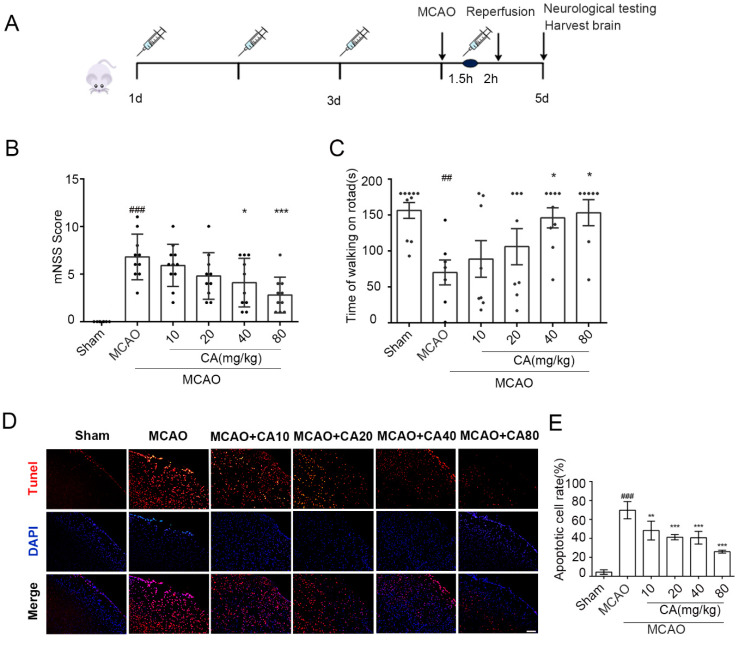
CA decreased cerebral damage and apoptosis following ischemia-reperfusion in vivo. (**A**) The timeline of CA administration in the OGD/R model. (**B**,**C**) Mouse behavioral performance in the mNSS score test and rotarod test with different treatments. (**D**,**E**) Effect of CA on apoptosis on the affected side of mice (TUNEL staining). Three animals were randomly selected from each group, and five fields of view were randomly selected from each section. ## *p* < 0.01, and ### *p* < 0.001 compared with the sham group. * *p* < 0.05, ** *p* < 0.01, and *** *p* < 0.001 versus MCAO group.

**Figure 5 nutrients-14-05390-f005:**
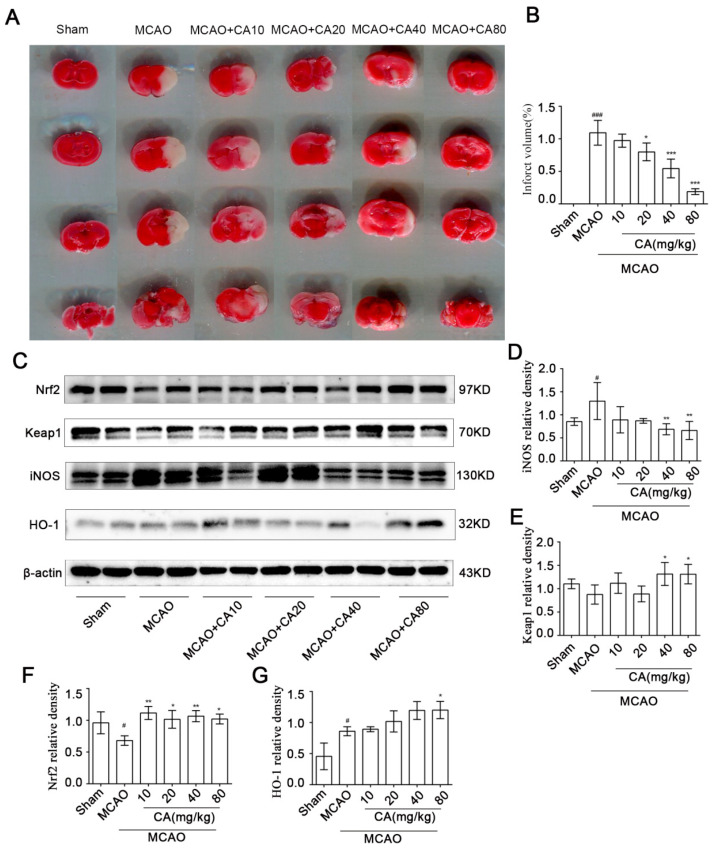
CA reduced infarct volume after ischemia-reperfusion in vivo. (**A**,**B**) Cerebral infarct volumes of rats in different treatment groups. Red staining represents normal tissues, and white staining represents the infarct region. (**C**–**G**) Chebulic acid increases activation of the Nrf2-ARE pathway in vivo. Western blot analysis was performed with specific antibodies to detect the activation of Nrf2, Keap1, iNOS, and HO−1 in the ischemic hemisphere (representative images from three independent experiments are shown). The results are expressed as mean ± SD. # *p* < 0.05 and ### *p* < 0.001 compared with the sham group. * *p* < 0.05, ** *p* < 0.01, and *** *p* < 0.001 versus MCAO group.

## Data Availability

Not applicable.

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
