# Peer review of "Chebulic Acid Prevents Hypoxia Insult via Nrf2/ARE Pathway in Ischemic Stroke"

_nutrients, 2022, doi:10.3390/nu14245390_

Round 1
Reviewer 1 Report
This research is very actual and up to date with robust research design planning and lab work done.
Major comments
1. CA mechanism of action is expected to be polemicizing and discussed in a more detailed way.
2. SH-SY5Y cells – glioblastoma cells – why this cell line was chosen? It is not obvious how this cell line can serve as the in vivo model of ischemic stroke.
3. Drug concentration choice – it is advised to explain 10 mg/kg vs 20 mg/kg vs 40 mg/kg vs 80 mg/kg dosage choice. What is the rationale behind these doses?
Minor comments
1. Line 27 “Enormous” potential – it might be recommended to use more modest phrasing for this, like “promising”
Reviewer 2 Report
The authors have done a detailed and thorough study on the beneficial effects of Chebulic acid(CA) in preventing hypoxia insult in ischemic stroke. The research methodology is well planned and the article fits the scope of the journal. However, there are few questions/changes that needs to be addressed before the manuscript can be considered for publication.
Major concerns/revisions:
1. Results section 3.3: The data in figure 3 seems to be contradictory. In 3A to 3D the authors have stated that CA enhances the translocation of Nrf2 to the nucleus which in turn activates the Nrf2-ARE signaling. However, in the figure legend - line 291 states ‘Chebulinic acid prevented an increase in the activation of Nrf2-ARE pathways’ while the title of the section 3.3 states ‘CA promotes nuclear translocation of Nrf2 in SH-SY5Y cells induced by OGD/R’; both of which are contrary. Please explain the rationale or correct the error accordingly.
2. Also, in the lower panel 3E-H the authors have shown the expression of Keap1 to be increased significantly in the 40 mg/ml CA group when compared to the control. It is well known that Keap1 is a negative regulator of Nrf2-ARE activation as it sequesters Nrf2 in the cytoplasm and directs it for Cullin-3 mediated ubiquitination and degradation. Additionally, there is plenty of evidence that increased expression of Keap1 downregulates Nrf2-ARE pathway or in other words Keap 1 deficiency facilitates Nrf2 translocation and ARE binding (PMID- 23867319; 34063933). Please elucidate how the increase in the expression of Keap1 activates the translocation of Nrf2 to the nucleus and enhances the Nrf2-ARE signaling. The same question applies to the in vivo work as well (Figure 5).
3. In figure 5 panel G (HO-1 bar graph), the bars/values for CA 40mg/kg and 80 mg/kg look alike. However, there is no significance for the 40 mg group when compared to the MCAO while the difference is statistically significant for the 80mg group?
Minor corrections:
1. Abstract: Lines 16 & 17- Please change the sentence as ‘In this study, we investigated the mechanism of anti-hypoxic neuroprotective effect of CA in vitro and in vivo’.
2. Methods: Lines 75 & 76- the reviewer suggests to remove the phrase ‘CA was dissolved in glucose-free DMEM at different concentrations,’ as it doesn’t make sense to include it in that paragraph.
3. Line 96- The sentence ‘Cells were inoculated in 96-well plates (104 ml).’ is repeated, kindly remove.
4. Section 2.13, Line 201- Is the protein load 30 mg per lane or 30 mg per lane?
5. Line 333-figure legend: ‘The results are expressed as ……. ± S.E.M.’ Please mention the measure (mean ±).
6. A pictorial or graphical representation of the proposed mechanism of action of CA on ischemic stroke could give more insights and a better understanding for the readers about the role of Nrf2-ARE pathway in culminating hypoxia insult during ischemic stroke.
7. Give a brief note in the introduction section about the role of oxidative stress in the pathogenesis of hypoxic insult and ischemic stroke and also how activation of Nrf2-ARE signaling could culminate it.
8. It is suggested that the references be numbered in the order of appearance in the text (example reference number 12-13 appear before number 10 and 11), unless there is a specific reason for the irregularity in numbering.
9. Are there any known side effects/toxicology data from rodent or human studies available for CA?
Reviewer 3 Report
In this manuscript, the authors explore the neuroprotective effect of a chinese medicine Chebulic acid, in ischemia reperfusion injury. CA is known for its antioxidant and anti-cancer properties.
The authors measure the nuclear translocation of Nrf2 and reactive oxygen species, to test for the antioxidant properties. They perform in vitro experiments using an SHSY5Y cells OGD/R model and subject the cells to various concentrations of CA. The authors report a dose-dependent inhibition in ROS formation and reduction in apoptotic cells, in addition to nuclear translocation of Nrf2. They investigate the effect of the drug in multiple genes in the Nrf2 pathway. They validate their results in a middle cerebral artery occlusion mouse model and observe an improvement in behavior in a rotarod test. They conclude that the CA shows neuroprotective effects in an MCAO model.
-Overall, the authors suggest neuroprotective effects of CA In I/R mouse model and their experimental results validate their findings.
The Methods section needs some verification and changes.
-the authors use a stock of 60µg/ml, is there a typographical error in the drug concentration
-There needs some verification in the immunofluorescence and detection of the ROS and apoptosis rate by flow cytometry; There is a repetition and mix up of titles.
-The authors report nuclear Nrf2 expression in western blots, was this done by separating nuclear fraction? The authors need to clarify this.
-The figure 5C shows duplicate bands for each condition, does this represent both hemispheres or is this duplicate, can the authors clarify this in the legends.
-Did the authors observe any side effects of the drug on other organs in vivo.
